# Assessing the Respiratory Effects of Air Pollution from Biomass Cookstoves on Pregnant Women in Rural India

**DOI:** 10.3390/ijerph18010183

**Published:** 2020-12-29

**Authors:** Raj Parikh, Sowmya R. Rao, Rakesh Kukde, George T. O’Connor, Archana Patel, Patricia L. Hibberd

**Affiliations:** 1Division of Pulmonary and Critical Care Medicine, Boston University School of Medicine, Boston University Medical Center 72 East Concord Street, R304, Boston, MA 02118, USA; goconnor@bu.edu; 2Department of Global Health, Boston University School of Public Health, Boston, MA 02118, USA; srrao@bu.edu (S.R.R.); plh0@bu.edu (P.L.H.); 3Lata Medical Research Foundation, Nagpur 440002, India; rajahal3@gmail.com (R.K.); dr_apatel@yahoo.com (A.P.)

**Keywords:** household air pollution, biomass fuel, fractional exhaled nitric oxide, Central India, indoor air pollution

## Abstract

Background: In India, biomass fuel is burned in many homes under inefficient conditions, leading to a complex milieu of particulate matter and environmental toxins known as household air pollution (HAP). Pregnant women are particularly vulnerable as they and their fetus may suffer from adverse consequences of HAP. Fractional exhaled nitric oxide (FeNO) is a noninvasive, underutilized tool that can serve as a surrogate for airway inflammation. We evaluated the prevalence of respiratory illness, using pulmonary questionnaires and FeNO measurements, among pregnant women in rural India who utilize biomass fuel as a source of energy within their home. Methods: We prospectively studied 60 pregnant women in their 1st and 2nd trimester residing in villages near Nagpur, Central India. We measured FeNO levels in parts per billion (ppb), St. George’s Respiratory Questionnaire (SGRQ-C) scores, and the Modified Medical Research Council (mMRC) Dyspnea Scale. We evaluated the difference in the outcome distributions between women using biomass fuels and those using liquefied petroleum gas (LPG) using two-tailed t-tests. Results: Sixty-five subjects (32 in Biomass households; 28 in LPG households; 5 unable to complete) were enrolled in the study. Age, education level, and second-hand smoke exposure were comparable between both groups. FeNO levels were higher in the Biomass vs. LPG group (25.4 ppb vs. 8.6 ppb; *p*-value = 0.001). There was a difference in mean composite SGRQ-C score (27.1 Biomass vs. 10.8 LPG; *p*-value < 0.001) including three subtotal scores for Symptoms (47.0 Biomass vs. 20.2 LPG; *p*-value< 0.001), Activity (36.4 Biomass vs. 16.5 LPG; *p*-value < 0.001) and Impact (15.9 Biomass vs. 5.2 LPG; *p*-value < 0.001). The mMRC Dyspnea Scale was higher in the Biomass vs. LPG group as well (2.9 vs. 0.5; *p* < 0.001). Conclusion: Increased FeNO levels and higher dyspnea scores in biomass-fuel-exposed subjects confirm the adverse respiratory effects of this exposure during pregnancy. More so, FeNO may be a useful, noninvasive biomarker of inflammation that can help better understand the physiologic effects of biomass smoke on pregnant women. In the future, larger studies are needed to characterize the utility of FeNO in a population exposed to HAP.

## 1. Introduction

An estimated half of the world’s population, predominantly in low–middle income countries, utilizes biomass fuels as their main source of energy for cooking, heating, and lighting [1,2]. Biomass fuels such as woody fuels and animal wastes are organic materials. Biomass fuel is burned in the home under inefficient conditions, leading to a complex milieu of particulate matter and environmental toxins known as household air pollution (HAP). HAP, despite being a modifiable and preventable risk factor in the global burden of disease, has been identified by the WHO as a major cause of morbidity worldwide behind unsafe drinking water and sanitation [3,4,5]. In 2018, the WHO estimated that 3 billion people cook using biomass fuels [3]. Furthermore, almost 4 million people die from HAP annually as a result of pneumonia, chronic obstructive lung disease (COPD), lung cancer, and other medical complications [3].

Due to time spent in the household, pregnant women are particularly vulnerable to HAP [6,7]. They and their fetus may suffer adverse consequences of exposure to biomass smoke [8,9,10,11,12,13,14,15]. For these women, exposure to biomass smoke increases the risk of anemia, cardiovascular disease, and perinatal complications [10,16]. The toxic air particles can not only cause deficits in lung function but also cross the placenta and reduce oxygen delivery to the fetus [11,17,18]. Increased levels of these pollutant components result in decreased birth weight, height, and head circumference while also serving as a risk factor for preterm birth [11,17,19,20,21,22,23]. Carbon monoxide (CO), in particular, is a health-damaging airborne pollutant that can reduce oxygen delivery to the fetal circulation thereby causing tissue hypoxia [24,25,26,27].

The underlying mechanisms of pulmonary disease as a consequence of air pollution are believed to be directly related to airway inflammation [28,29,30,31,32]. Established evidence suggests that an environmental exposure may result in increased reactive oxygen species and an upregulation of proinflammatory cytokines [33,34,35]. In order to assess this subclinical airway inflammation, fractional exhaled nitric oxide (FeNO) has recently been utilized in air pollution studies [33,36,37,38,39,40,41,42,43]. FeNO measurements quantify the amount of nitric oxide (NO) in one’s exhaled breath to indicate eosinophilic inflammation as a result of cytokines and Type 2 helper cells (Th2) [44,45,46,47]. As detailed by Allen et al., FeNO has been studied extensively in asthmatics as well as in COPD, cystic fibrosis, and bronchiectasis and is often referred to as a surrogate for airway inflammation [48].

Due to the deleterious effects of HAP in both the adult and pediatric population, the Indian government initiated a scheme in 2015 to provide subsidized clean fuel to those in the community who could not afford it. Liquefied petroleum gas (LPG), a mixture of propane and butane, is a clean and efficient fuel that the Indian government has utilized as a versatile energy source for those living below the poverty line. However, despite the subsidization of LPG, many households continue to use biomass fuel as a primary source of energy [49].

The objective of our study was to assess the burden of respiratory symptoms and airway inflammation, measured by FeNO levels, among pregnant women in rural India who utilize biomass fuel within their home, irrespective of individual cookstove design. We assessed whether the distribution of (i) FeNO measurements, (ii) self-reported St. George’s Respiratory Questionnaire (SGRQ-C) scores, and (iii) Modified Medical Research Council (mMRC) Dyspnea Scale varied substantially between women who lived in households using biomass fuel for cooking and those who used only LPG. A secondary objective was to assess the relationship of FeNO measurements with mMRC scores.

## 2. Materials and Methods

We conducted a cross-sectional study in rural villages near Nagpur, India. The study protocol was approved by the Institutional Review Boards of Boston University Medical Center and Lata Medical Research Foundation (H-38568). All study participants provided informed consent.

### 2.1. Study Setting

The study was conducted at government-established primary health centers (PHC) in seven rural villages within a 100-km radius of Nagpur, India. Nagpur is home to approximately 2.4 million people, where approximately 36% of the population lives in low-income communities.

### 2.2. Participants

Pregnant women within the first and second trimester of pregnancy, as defined by last menstrual period, were included in the study. Those in their third trimester were excluded as there is limited evidence regarding the physiologic changes of pregnancy that are heightened in the third trimester and its effects on FeNO testing. Subjects were considered ineligible if they reported a prior history of cardiopulmonary disease, were not planning to have all of their antenatal care in the Nagpur area, or were smoking tobacco-related products. The 65 women enrolled in the study were stratified into two primary groups based on the self-reported type of cooking fuel used in the household: (i) mainly biomass; (ii) only LPG. Information regarding type of cookstove being used, other than the primary fuel source, was not recorded.

### 2.3. FeNO Testing

FeNO testing with NIOX VERO (NIOX, Circassia, UK) was conducted during each visit. Participants sat upright, emptied their lungs, inhaled steadily through the NIOX VERO, and then exhaled at a slow and steady rate for 10 s. Each study participant performed two tests and the results of which were deemed satisfactory by the trained physicians conducting the test; the final FeNO measurement was the average of both tests. Satisfactory scores were defined as those that resulted in appropriate exhalation waveforms that triggered the device to provide a FeNO value, measured as parts per billion (ppb). FeNO testing was conducted by a trained pulmonary physician from Boston University Medical Center and a trained cardiology physician employed by the Lata Medical Research Foundation.

### 2.4. Subjective Testing

The SGRQ was originally designed to measure health impairment in patients with asthma and COPD. This respiratory questionnaire has been previously utilized and proven successful in low-resource areas of India [50,51,52]. The SGRQ-C is a shorter questionnaire derived from the original version following detailed data analysis of large studies in COPD [53]. The SGRQ-C was adapted for our study in an effort to obtain a subjective assessment of each participants’ respiratory status. The questionnaire included 14 prompts (9 multiple choice questions and 5 true versus false questions) and the first section focused on respiratory symptoms such as cough, phlegm production, shortness of breath, and wheezing. Two additional sections assessed the subjects’ quality of life and exercise capacity as it pertained to their respiratory status. The SGRQ-C was translated into Hindi and Marathi by the publisher and was completed by each participant individually. The SGRQ-C scoring tool, provided by the publisher, was used to create four scores: Symptoms, Activity, Impact, and Composite. The adjusted scores were computed based on the methodology provided by St. George’s University of London and the higher scores indicate worse functionality [54]. The mMRC scale was also used as a second subjective assessment of the participants’ symptoms [55]. The mMRC scale is a self-rating tool that is used to measure the degree of dyspnea that affects daily activities. The scale ranges from 0 to 4 with the higher number implying more severe respiratory symptoms.

### 2.5. Covariates

Age, body mass index (BMI) recorded at initial prenatal visit, level of education, prior pregnancy history, location of cookstove within or outside the home, and second-hand smoking status were recorded for each woman participating in the study. This was completed using an interviewer-administered questionnaire designed specifically for the study.

### 2.6. Statistical Analysis

Summary statistics (e.g., proportion, mean, standard deviation (SD)) were obtained on all variables. As appropriate, statistical significance of group differences were tested with Fisher’s exact tests (categorical variables) and Kruskal-Wallis tests (continuous variables). Pearson correlations were used to quantify the linear relationship between Mean FeNO and mMRC. All analyses were performed using SAS 9.4 (SAS Institute, Cary, NC, USA) with a two-sided *p* < 0.05 considered statistically significant.

## 3. Results

Of the 110 subjects prescreened, 65 eligible subjects were enrolled (Figure 1). We excluded 5 subjects (3 in the biomass group, 2 in the LPG group) unable to follow instructions for FeNO testing. Of the 60 subjects, 28 lived in households using only LPG, 9 in households using only Biomass, and 23 in households using Biomass with LPG. We combined the last two groups since the distribution of the measurements in these groups was similar; this combined group was referred to as Biomass while the only LPG group was referred to as LPG.

The distribution of baseline participant characteristics including age, BMI, level of education, and second-hand smoke exposure were similar in the LPG and Biomass groups (Table 1). Of note, 40.6% (n = 13) of women in the Biomass group had a history of preterm birth compared to only 7.1% (n = 2) of women in the LPG group, similar to findings previously well-established by our co-investigators [8]. The majority of participants reported that their cookstove was physically located within the home (84.4% in Biomass; 96.4% in LPG) and not in a separate building. Among those using biomass, wood was the most common source of biomass fuel (75.0%).

As mentioned above, FeNO testing was conducted for each participant until two satisfactory scores were obtained with appropriate exhalation curves. Multiple attempts at obtaining FeNO measurements were required for 47% of the Biomass group and 53.6% of the LPG group. The mean FeNO score for the Biomass group was 25.4 ppb compared to 8.6 ppb in the LPG subgroup (*p* < 0.001; Table 2).

The SGRQ-C assessed three primary groups of respiratory health, based on the participants’ symptoms, activity tolerance, and impact or quality of life. On average, the Biomass group had higher adjusted scores than the LPG group (Symptom: 47.0 vs. 20.2; Activity: 36.4 vs. 16.5; Impact: 15.9 vs. 5.2; Composite: 27.1 vs. 10.8; all *p* < 0.0001; Table 2). In particular, the Biomass group reported more symptoms (Table 3 and Table 4) than the LPG group for cough (41.6 vs. 21.0; *p* < 0.0001; Table 4), phlegm production (44.4 vs. 20.7; *p* < 0.001), and shortness of breath (46.3 vs. 18.0; *p* < 0.001). Both groups scored similarly for wheezing (5.7 in Biomass vs. 5.2 in LPG; *p* = 0.89).

The mean mmRC score from the Biomass group was 2.9 compared to 0.5 in the LPG group (*p* < 0.001; Table 2). Results of the secondary analysis indicated that FeNO measurements were highly correlated with mMRC scores (r = 0.80; Figure 2).

## 4. Discussion

Our cross-sectional study demonstrated increased prevalence of respiratory symptoms and airway inflammation, as measured by FeNO, among pregnant women using biomass fuel compared to those using LPG. Our study also provided a unique opportunity to evaluate correlations between the subjective respiratory assessment of dyspnea and FeNO levels.

### 4.1. Respiratory Symptoms

Prior studies evaluating the burden of respiratory symptoms among those exposed to biomass fuel have adapted various methodologies. A recent investigation into biomass fuel exposure by Pathak et al. adapted its own questionnaire designed specifically for the study, focusing on common respiratory symptoms such as cough, phlegm production, wheezing, and dyspnea [56]. While there remains significant overlap with validated questionnaires, this method lacks reproducibility in other, similarly framed studies. Meanwhile, the SGRQ has been utilized not only in India but also in other countries, including rural Bolivia, as a valuable tool to analyze the respiratory effects of HAP [57]. Our study adopted a two-prong method of assessing subjective symptoms using both a self-administered questionnaire and interviewer-administered dyspnea scale. This approach, which was utilized by Kurmi et al. in a biomass exposure study in Nepal, has a clear advantage in possibly avoiding challenges faced when using a translated questionnaire in a different cultural setting [58,59]. Not only did we utilize both the SGRQ-C and mMRC, but our study also found a correlation between subjective and objective measurements of mMRC and FeNO, respectively (Figure 2). This correlation works to expand on the concept that respiratory symptoms alone may be early indicators of chronic airway disease and early identification of these symptoms would allow for preventative care and intervention [60,61].

### 4.2. FeNO

The evidence from our study supports the hypothesis that increased exposure to HAP is linked to airway inflammation as measured by FeNO. FeNO provides a noninvasive, safe, and accurate method for assessing the degree of airway inflammation in patients [62]. However, prior studies have shown inconsistent findings: Benka-Coker et al. failed to show a connection between biomass cookstoves and FeNO measurements among Honduras women [63]. Similar null associations were seen in two other studies by Strak et al. and Yoda et al. [64,65]. Meanwhile, Berhane et al. and Liu et al., among others, have shown the utility of FeNO in assessing air pollution effects in children [40,41,42,66,67]. More recently, Tamasi et al. examined the role of FeNO in adults, including pregnant women and several additional investigations have followed course [33,36,37,38,39,43]. The reason for the inconsistent data regarding FeNO could be related to the technology available to assess exhaled NO. In particular, Benka-Coker et al. hypothesized that optimizing the flow rate within the device may provide a more accurate assessment of the proximal airways thereby increasing the accuracy of FeNO testing [63].

Our study contributes to the growing body of evidence that FeNO can be a useful tool in assessing the consequences of biomass fuel exposure on the respiratory system and thereby provide insight into the physiologic effects of toxic inhalation on the airways. Malerba et al. showed increased levels of NO synthase in airway epithelial cells of patients with asthma as well as reduced levels in those who are treated with inhaled corticosteroids [68]. Similarly, Olin et al. established that increasing values of FeNO are seen in those with atopic disease, with or without respiratory complaints [69]. Both Erpenbeck et al. and Brightling et al. have indicated a positive correlation between FeNO levels and eosinophil count on bronchoalveolar lavage fluid [68,69]. These prior investigations, in conjunction with the findings from our study, can help us better understand the physiologic changes that are taking place in the respiratory system as a result of biomass fuel exposure.

Given the emerging use of FeNO in air pollution studies on a global level as well as the results from our investigation where we established a correlation between mMRC and FeNO measurements, this tool shows great promise in helping identify those who have developed a degree of respiratory compromise as a result of HAP exposure.

### 4.3. Strengths

A major strength of our study is the comprehensive respiratory assessment of participants. All subjects completed a self-administered questionnaire, an interviewer-administered assessment of dyspnea, and an objective test to measure airway inflammation. This allows for a precise evaluation of the impact that HAP has on the pulmonary function of the subjects.

### 4.4. Limitations

Limitations of this study, first and foremost, arise from the fact that we did not directly measure pollutant concentrations and instead, used self-reported proxy for fuel use. Another factor was that we did not have information regarding each study participants’ type of cookstove within the home. The effects of outdoor air pollution along with second-hand smoke exposure could have been a confounding factor as well but we did not have information that we could adjust for. The cross-sectional nature of the study design limits our ability to establish whether the exposure to biomass fuel preceded the evidence of airway inflammation. Lastly, the small sample size of our study is another limitation and could prevent the findings from being generalized to larger populations that are exposed to biomass fuel.

## 5. Conclusions

Reliance on biomass fuels as a primary source of household energy and the resultant HAP, which is a modifiable risk factor, contribute directly to the high burden of respiratory disease in India. Elevated FeNO levels and higher dyspnea scores in biomass-fuel-exposed subjects confirm the adverse respiratory effects of this exposure during pregnancy, adding to previous findings among similar group of women. More so, FeNO may be a useful, noninvasive biomarker of inflammation that can help better understand the physiologic effects of biomass smoke on pregnant women. Further research is needed to characterize the physiologic utility of FeNO in a population exposed to HAP as a means of developing cost-effective strategies aimed at improving respiratory health.

## Figures and Tables

**Figure 1 ijerph-18-00183-f001:**
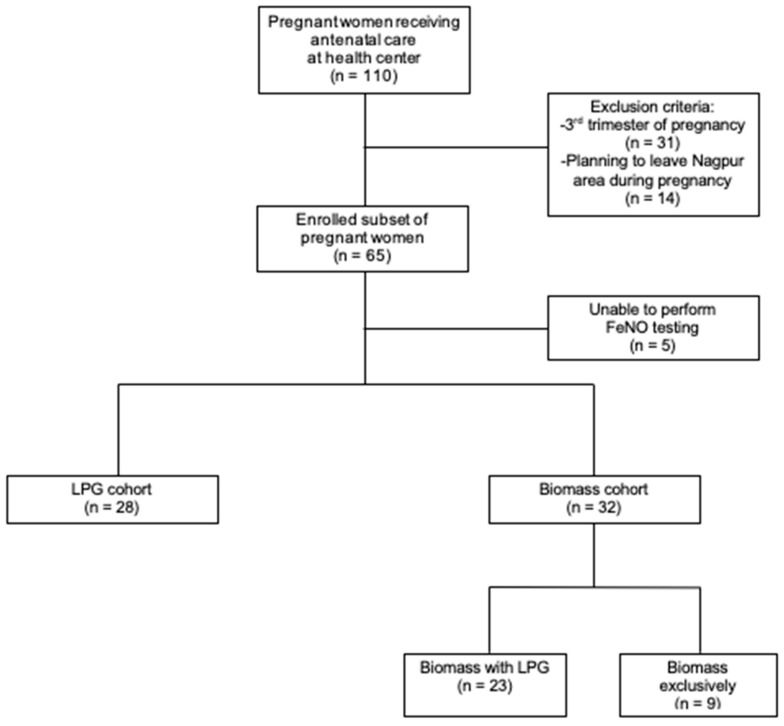
Consort diagram.

**Figure 2 ijerph-18-00183-f002:**
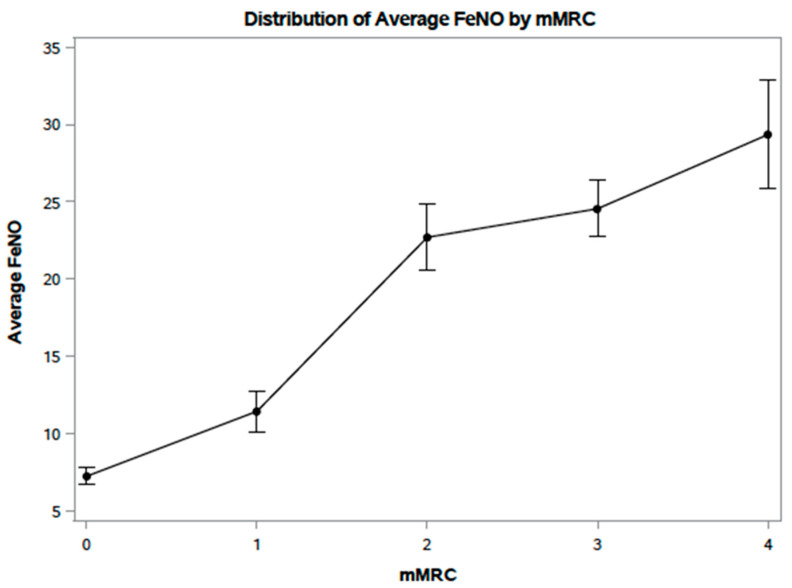
Scatter plot between FeNO results and mMRC dyspnea scale scores.

**Table 1 ijerph-18-00183-t001:** Participant characteristics. LPG: liquefied petroleum gas.

Participant Characteristics	Biomass (n = 32)	LPG (n = 28)	*p*-Value
Mean age (SD)	23.3 (3.0)	23.4 (3.1)	0.987
Mean BMI (SD)	19.7 (2.1)	18.9 (1.6)	0.109
Education			1.0
Primary	9 (28.1%)	8 (28.6%)	
Secondary	12 (37.5%)	10 (35.7%)	
College	11 (34.4%)	10 (35.7%)	
Parity			0.802
Nulliparous	16 (50.0%)	15 (53.6%)	
Multiparous	16 (50.0%)	13 (46.4%)	
Preterm birth history	13 (40.6%)	2 (7.1%)	0.003
Second hand smoke			0.946
Daily	15 (46.9%)	13 (46.4%)	
Weekly/Monthly	8 (25.0%)	6 (21.4%)	
Never	9 (28.1%)	9 (32.1%)	
Biomass fuel type		Not applicable	
Wood	24 (75.0%)		
Crop/grass	3 (9.4%)		
Cow dung	5 (15.6%)		
Secondary fuel type		Not applicable	
Crop/grass	22 (68.8%)		
Cow dung	3 (9.4%)		
Fuel source timeframe			<0.001
<3 years	4 (12.5%)	18 (64.3%)	
≥3 years	28 (87.5%)	10 (35.7%)	
Stove location within home			0.201
Within home	27 (84.4%)	27 (96.4%)	
Separate room	5 (15.6%)	1 (3.6%)	

**Table 2 ijerph-18-00183-t002:** St. George’s Respiratory Questionnaire (SGRQ-C) scores, Modified Medical Research Council (mMRC) dyspnea scale, and mean fractional exhaled nitric oxide (FeNO) results between the Biomass and LPG groups of the study.

SGRQ-C, mMRC, FeNO Scores	Biomass (n = 32)	LPG (n = 28)	*p*-Value
SGRQ-C Symptoms	47.0 (5.5)	20.2 (11.1)	<0.001
SGRQ-C Activity	36.4 (8.5)	16.5 (9.0)	<0.001
SGRQ-C Impact	15.9 (4.7)	5.2 (3.5)	<0.001
SGRQ-C Composite	27.1 (3.8)	10.8 (4.0)	<0.001
Mean mMRC	2.9 (0.9)	0.5 (0.6)	<0.001
FeNO (first test) ppb	25.4 (7.9)	8.6 (3.2)	<0.001
FeNO (second test) ppb	25.4 (8.1)	8.6 (3.1)	<0.001

**Table 3 ijerph-18-00183-t003:** Symptoms section of the SGRQ-C including comparisons between Biomass and LPG group for Cough, Phlegm production, Shortness of breath, and Wheezing.

SGRQ-C Symptoms	Biomass (n = 32)	LPG (n = 28)
Cough		
Most days	5 (15.6)	0 (0.0)
Several days	14 (43.8)	3 (10.7)
Only with chest infections	10 (31.3)	16 (57.1)
Not at all	3 (9.4)	9 (32.1)
Phlegm		
Most days	4 (12.5)	0 (0.0)
Several days	16 (50.0)	4 (14.3)
Only with chest infections	12 (37.5)	13 (46.4)
Not at all	0 (0.0)	11 (39.3)
Shortness of breath		
Most days	2 (6.3)	0 (0.0)
Several days	26 (81.3)	10 (35.7)
Not at all	4 (12.5)	18 (64.3)
Wheezing		
Most days	0 (0.0)	0 (0.0)
Several days	0 (0.0)	0 (0.0)
Few days	0 (0.0)	0 (0.0)
Only with chest infections	5 (15.6)	4 (14.3)
Not at all	27 (84.4)	24 (85.7)

**Table 4 ijerph-18-00183-t004:** Weighted scores for the SGRQ-C Symptoms section.

Weighted Scores	Biomass (n = 32)	LPG (n = 28)	*p*-Value
Cough	41.6 (21.9)	21.0 (15.7)	<0.001
Phlegm	44.4 (14.7)	20.7 (17.9)	<0.001
Shortness of breath	46.3 (19.9)	18.0 (24.5)	<0.001
Wheezing	5.7 (13.4)	5.2 (13.0)	0.887

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
