# Peer review of "Assessing the Respiratory Effects of Air Pollution from Biomass Cookstoves on Pregnant Women in Rural India"

_ijerph, 2020, doi:10.3390/ijerph18010183_

Round 1
Reviewer 1 Report
Summary
In this study, the effects of household use of biomass fuels on the respiratory health of pregnant women in India was investigated. The sample group included 32 women from households mainly using biomass and 28 women from households exclusively using liquid petroleum gas (LPG) in villages near Nagpur, India. Respiratory health was measured through three separate tests: measurement of FeNO levels in exhaled air as an indication of airway inflammation; the St. George’s Respiratory Questionnaire; and the Modified Medical Research Council Dyspnea Scale. The differences in outcome between the two groups of women were evaluated with two-tailed t-tests. It was found that, while the demographics of the two groups of pregnant women were similar, the women exposed to biomass burning emissions have higher FeNO levels, higher symptom, activity and impact SGRQ-C scores, and higher dyspnea scores. It was concluded that the household use of biomass fuels (rather than a cleaner energy source) negatively affects the respiratory health of pregnant women, and that FeNO is a useful indicator of airway inflammation that may well be correlated with other negative respiratory effects.
Overall, I feel that this study is succinctly written and clearly presented. The results are robust because three separate approaches of evaluating respiratory disease were followed. The limitations of the study are clearly acknowledged.
Major Issues
There are no major issues to be addressed in this paper before publication.
Minor Issues
Please address the following:
- Line 168-171: It is not clear what the figures in brackets refer to in these lines, since the values do not correspond with the values in Table 3.
- Tables 2, 3, 4: Please provide more comprehensive titles for the tables and indicate what the values in brackets are.
- Figure 2: Please also provide a more comprehensive caption for the figure.
- Section 4.4: I think the high incidence of daily exposure to second-hand smoke and the lack of information on the contribution that this makes to HAP is another limitation of the study.
Author Response
- Line 168-171: It is not clear what the figures in brackets refer to in these lines, since the values do not correspond with the values in Table 3. This is a great point and the incorrect Table was originally being referenced. We have adjusted the text to relay that Table 4, not Table 3, has those values listed from the text.
- Tables 2, 3, 4: Please provide more comprehensive titles for the tables and indicate what the values in brackets are. We have provided more detailed titles for each table from 2, 3, and 4 that better highlights what is being listed and covered in regards to data from each table.
- Figure 2: Please also provide a more comprehensive caption for the figure. Similar to prompt 2, we have provided a better detailed description for the caption of Figure 2.
- Section 4.4: I think the high incidence of daily exposure to second-hand smoke and the lack of information on the contribution that this makes to HAP is another limitation of the study. We have included second-hand smoke exposure in the list of limitations for the study, as detailed in the 4.4 Limitations section.
Reviewer 2 Report
Reviewer
Abstract:
Methods: We prospectively studied 60 pregnant women in their 1st and 2nd trimester residing in 20 villages near Nagpur, Central India.
Results: Sixty subjects (32 in Biomass households; 28 in LPG households) were enrolled in the study.
It is suitable
Introduction
It is suitable
Materials and Methods
Participants
Lines 99 – 100 : The 65 women enrolled in the study were stratified into two primary groups based on the self-reported type of cooking fuel used in the household……
In the abstract it is placed in methods
Abstract: Methods: We prospectively studied 60 pregnant women in their 1st and 2nd trimester residing in 20 villages near Nagpur, Central India.
Are there 60 or 65 participants?
Statistical methods
It is suitable
Results
Biomass
n=32
LPG
n=28
It is suitable
Results
Line 142 - Of the 110 subjects pre-screened, 65 eligible subjects were enrolled (Figure 1).
In the abstract and in methods there are 60 or 65 and in the results there are 110 participants.
Please explain further.
Discussion
Limitations
Limitations of this study, first and foremost, arise from the fact that we did not directly measure pollutant concentrations and instead, used self-reported proxy for fuel use.
Despite this limitation the discussion is suitable
Conclusion
It is suitable
References
It is suitable
Thank you
Author Response
Abstract:
Lines 99 – 100 : The 65 women enrolled in the study were stratified into two primary groups based on the self-reported type of cooking fuel used in the household……
In the abstract it is placed in methods
Abstract: Methods: We prospectively studied 60 pregnant women in their 1st and 2nd trimester residing in 20 villages near Nagpur, Central India.
Are there 60 or 65 participants?
This has been addressed in the Abstract. The study enrolled 65 subjects but only 60 completed the study and 5 were excluded after not being able to perform FeNO testing.
Results
Line 142 - Of the 110 subjects pre-screened, 65 eligible subjects were enrolled (Figure 1).
In the abstract and in methods there are 60 or 65 and in the results there are 110 participants.
Please explain further.
This has been addressed in the Abstract. The study enrolled 65 subjects but only 60 completed the study and 5 were excluded after not being able to perform FeNO testing.
Reviewer 3 Report
Major points
This is an interesting paper on an important subject. I liked the use of complementary methods, even if I have questions regarding one of these methods (attribution of measured FeNO to airway inflammation).
The assumption in the paper appears to be that the presence of NO in exhaled air is a consequence only of respiratory distress. No mention is made in the paper of the production of NO as a product of combustion, or of how it is accounted for. I would expect combustion NO to be in concentrations in the same order as that reported for FeNO in the paper, so it is unclear to what extent the FeNO results are a sign of airway inflammation or differences in NO production from the burning of the different fuels. I have given "Low" for scientific soundness because without knowing whether the NO measurements reflect combustion products or airway inflammation it is very difficult to judge the merits of that part of the paper.
Further details of the groups would be useful with respect to:
- How long had the LNG users been using LNG?
- Was there variation in the types of stove being used – not all biomass stoves are equal?
- Were there differences in the ventilation of households – particularly, did they all have chimneys to remove combustion products (if the biomass stoves were unvented and the LNG stoves were, the study may not be showing a difference between LNG and biomass, but between venting and not)?
- Were there differences in the location of households? Perhaps the biomass houses tended to be in rural areas of lower population density.
Some of these issues are important in determining what is responsible for the differences in reported symptoms.
Perhaps this is outside the expertise of the authors, but a comment about the efficacy of the policy of promoting LNG seems worthwhile, when many households don't make the switch that the policy seems to aim at, but burn both biomass and LNG.
Minor points
Line 13: implies a generality that may not be true.
Line 43: Not all of these fuels can be considered ‘renewable’
Line 45: Studies are old and could be updated with new results from Global Burden of Disease
Paragraphs starting lines 53 and 62: The concentrations associated with harm, and their relationship with concentrations in typical Indian homes would be useful added information.
Line 146: Unclear what characteristics are referred to here.
Author Response
- We have provided information regarding the “fuel source timeframe”. What was meant by this data point was what the subjects reported was the timeframe of their use of the reported fuel, whether it be LPG or Biomass. We divided this into less than 3 years versus greater than or equal to 3 years. In the LPG group of 28 subjects, 18 subjects (64.3%) reported use of LPG for less than 3 years whereas 10 subjects (35.7%) reported LPG use for greater than or equal to 3 years.
- This is a great point and a very valid and relevant one in regards to our study. The reviewer is absolutely correct in noting that there are multiple types of biomass stoves and they are not all made “equal”. However, we were not able to delineate information regarding each type of cookstove being used by the subjects. This is a limitation to our study and something we will keep in mind when constructing offshoot studies in the near future.
- This is also a very great point made by the reviewer. We felt it was important to have a decent understanding of the subjects’ household situation as it related to the cookstove. If one house had better venting than the other, perhaps that was the reason for pulmonary compromise and not necessarily the type of fuel being used, as the reviewer is implying. While we did not include this information within our study, there were no subjects who had homes with chimneys in our study. Homes in the areas where this study took place included subjects who did not have chimneys and that can be definitely stated. Similarly, and what we included in our study, was the stove location within the home and how that may relate to appropriate ventilation.
- All of the houses where subjects lived were in a rural setting. There were no subjects who were enrolled in the study that lived in an urban setting. This is stated at the start of Section 2 where it is written that “rural villages” near Nagpur, India is where the study was conducted.
Minor points
Line 13: implies a generality that may not be true. We have added the word “many” to imply that not EVERY home burns biomass fuel but that several, in fact, do.
Line 43: Not all of these fuels can be considered ‘renewable’ We have deleted the term renewable in this sentence as the reviewer makes a great point to suggest that not all the fuels listed are technically renewable.
Line 45: Studies are old and could be updated with new results from Global Burden of Disease
Paragraphs starting lines 53 and 62: The concentrations associated with harm, and their relationship with concentrations in typical Indian homes would be useful added information. This is a great point and has been referenced at various other points in the manuscript. We respect and understand the reviewer’s point regarding newer references and information. The paragraph discussed here from lines 53 to 62 is important to highlight the vulnerable population of pregnant women by expanding on the pathophysiology of this vulnerability. That is the main focus of this paragraph at this point in the manuscript.
Line 146: Unclear what characteristics are referred to here. We have replaced the term characteristics with measurements, as this is more accurate and precise of a word that describes the reasoning behind combining the Biomass only and Biomass with LPG groups.
Round 2
Reviewer 3 Report
Major points
The authors do not appear to have added any information that tells the reader whether the increased exposure is attributable to the use of biomass vs LPG, or to differences in stove / ventilation design. Their reference (3) - Venkataraman et al 2010 concerns the Indian national initiative for advanced biomass cookstoves - the benefits of clean combustion. It is clearly relevant to the paper whether the study subjects (the pregnant women) were using advanced biomass stoves or the traditional, less efficient stoves.
If they did not collect this data there are still possibilities to properly reflect this in the paper:
- Go back and collect the data if possible.
- Change the title to "Assessing the Respiratory Effects of Air Pollution from Biomass Cookstoves on Pregnant Women in Rural India" and explicitly recognise that standards of stove design were not recorded in the paper. This recognition needs to go into the introduction, methods and discussion - for me it is a critical factor that has to be recognised, and a passing mention of it is insufficient.
The reason it is important is that the authors need to consider the policy consequences of their results. Should the priority be for more widespread rollout of LPG, or for improved stove design? have past government initiatives on both been a success or not?
Minor points
Introduction: The problem of old references remains. For example, the Global Burden of Disease / WHO studies have been repeatedly updated since 2004, and this needs to be recognised.
Also, it is stated that 780 million individuals rely on biomass for cooking, but the reference is from 2010 (which in turn suggests that the data used are older) equivalent to 65% of the 2010 population. Since 2010, the Indian economy has grown by between 5 and 8% annually, increasing overall by roughly 80%, the government introduced the LPG subsidy scheme in 2015 and there is a significant rate of urbanisation which will reduce access to biomass and improve access to other fuels. These are all factors that suggest to me that the 2010 reference cited is very much out of date. Again, this trend needs to be recognised, ideally with a more up to date reference.
Author Response
The authors do not appear to have added any information that tells the reader whether the increased exposure is attributable to the use of biomass vs LPG, or to differences in stove / ventilation design. Their reference (3) - Venkataraman et al 2010 concerns the Indian national initiative for advanced biomass cookstoves - the benefits of clean combustion. It is clearly relevant to the paper whether the study subjects (the pregnant women) were using advanced biomass stoves or the traditional, less efficient stoves.
If they did not collect this data there are still possibilities to properly reflect this in the paper:
- Go back and collect the data if possible.
- Change the title to "Assessing the Respiratory Effects of Air Pollution from Biomass Cookstoves on Pregnant Women in Rural India" and explicitly recognise that standards of stove design were not recorded in the paper. This recognition needs to go into the introduction, methods and discussion - for me it is a critical factor that has to be recognised, and a passing mention of it is insufficient.
The reason it is important is that the authors need to consider the policy consequences of their results. Should the priority be for more widespread rollout of LPG, or for improved stove design? have past government initiatives on both been a success or not?
We understand the reviewer’s point that there is an important distinction that must be made regarding differences in stove types and how this is a confounding factor in our study. Unfortunately, we do not have the ability to go back and collect this data. This is something we will prioritize as part of our research design in future adjunct studies. However, in regards to this manuscript, we will address this as a clearcut limitation and missing valuable piece of information. To start, we have edited the title of the paper as a whole as was suggested by the reviewer. Additionally, lines 79-80 in the Introduction, lines 103-104 in the Methods, and lines 245-246 in the Discussion. In the future, as is the implication in this study and astutely detailed by the reviewer, this will be a valuable piece of information to creating a stronger study design as a whole.
Minor points
Introduction: The problem of old references remains. For example, the Global Burden of Disease / WHO studies have been repeatedly updated since 2004, and this needs to be recognised.
Also, it is stated that 780 million individuals rely on biomass for cooking, but the reference is from 2010 (which in turn suggests that the data used are older) equivalent to 65% of the 2010 population. Since 2010, the Indian economy has grown by between 5 and 8% annually, increasing overall by roughly 80%, the government introduced the LPG subsidy scheme in 2015 and there is a significant rate of urbanisation which will reduce access to biomass and improve access to other fuels. These are all factors that suggest to me that the 2010 reference cited is very much out of date. Again, this trend needs to be recognised, ideally with a more up to date reference.
This is a very important point made by the reviewer and we respect the attention to detail that has been provided. The reviewer has gone through great lengths to help us optimize the overall quality of our research and our manuscript, and we truly appreciate that. To address the older statistics and older referenes, we have appropriately adjusted the formatting and the data provided in the first few paragraphs of the Introduction to the paper. We believe that with the more updated data, the text of the manuscript will be more reflective of the current statistics regarding biomass fuel and HAP. We are no longer using the outdated 2004 WHO data as can be seen in the revised manuscript.
